# Somatic Mutations Alter Interleukin Signaling Pathways in Grade II Invasive Breast Cancer Patients: An Egyptian Experience

**Auhood Nassar** [1,*], **Abdel Rahman N. Zekri** [1,*], **Mostafa H. Elberry** [1], **Ahmed M. Lymona** [2], **Mai M. Lotfy** [1], **Mohamed Abouelhoda** [3] and **Amira Salah El-Din Youssef** [1]

1 Cancer Biology Department, Virology and Immunology Unit, National Cancer Institute, Cairo University, Cairo 11796, Egypt
2 Surgical Oncology Department, National Cancer Institute, Cairo University, Cairo 11796, Egypt
3 Faculty of Engineering, Cairo University, Cairo 12613, Egypt
* Correspondence: auhood.nassar@nci.cu.edu.eg (A.N.); ncizekri@yahoo.com (A.R.N.Z.);
Tel.: +20-222-742-607 (A.N.)

**Abstract:** This study aimed to investigate the impact of somatic mutations on various interleukin signaling pathways associated with grade II invasive breast cancer (BC) in Egyptian patients to broaden our understanding of their role in promoting carcinogenesis. Fifty-five grade II invasive BC patients were included in this study. Data for somatic mutations in 45 BC patients were already available from a previous study. Data for somatic mutations of 10 new BC patients were included in the current study. Somatic mutations were identified using targeted next-generation sequencing (NGS) to study their involvement in interleukin signaling pathways. For pathway analysis, we used ingenuity variant analysis (IVA) to identify the most significantly altered pathways. We identified somatic mutations in components of the interleukin-2, interleukin-6, and inter-leukin-7 signaling pathways, including mutations in *JAK1, JAK2, JAK3, SOCS1, IL7R, MCL1, BCL2, MTOR,* and *IL6ST* genes. Interestingly, six mutations which were likely to be novel deleterious were identified: two in the *SCH1* gene, two in the *IL2* gene, and one in each of the *IL7R* and *JUN* genes. According to IVA analysis, interleukin 2, interleukin 6, and interleukin 7 signaling pathways were the most altered in 34.5%, 29%, and 23.6% of our BC group, respectively. Our multigene panel sequencing analysis reveals that our BC patients have altered interleukin signaling pathways. So, these results highlight the prominent role of interleukins in the carcinogenesis process and suggest its potential role as promising candidates for personalized therapy in Egyptian patients.

**Keywords:** multigene sequencing; breast cancer; Egyptian; somatic mutations; NGS; interleukin signaling pathways

## 1. Introduction

Breast cancer (BC) is the most common and first cause of cancer death among women [1]. In 2020, WHO reported 2.3 million women diagnosed with BC and 680,000 deaths worldwide [2]. In Egypt, it accounts for 33% of female cancers, and more than 22,000 new cases are diagnosed yearly [3]. There is a tight interplay between the immune system and the malignant cells. The resistance to the ability of the immune system to eliminate transformed cells and the presence of protumoral inflammation are two major contributing hallmarks of cancer [4]. In recent decades, chronic inflammation has been well established to have a critical role in tumor development and progression. It has proved to be one of the drivers of carcinogenesis in many cancers. Several studies have highlighted the key role of cytokines, especially interleukins (ILs), in BC initiation, migration, and progression through IL signaling in cancer cells [5,6]. A recent review by Manore et al. revealed that IL-6 is frequently activated in BC and simultaneously suppress the anti-tumor immune response. Moreover,

it highlighted that multiple IL-6 pathway components are promising therapeutic targets [7]. Another study showed that aberrant expression of IL-7 and its signaling intermediates in invasive BC could provide diagnostic and prognostic implications. In addition, measuring these molecules in breast tissues may provide molecular indicators of nodal status, tumor differentiation and aggressiveness [8]. In addition, other interleukins are well known to have function in BC occurrence as well as progression, such as: IL-1, IL-8, IL-11, and IL-12. It is well established that IL-1 levels are elevated in primary tumors or serum of BC patients. Additionally, it correlates with the aggressiveness of the disease phenotypes such as advanced stage, basal subtype, or metastasis [9]. On the other hand, numerous correlations between IL-8 expression in BC tumor cells and IL-8 plasma levels, and bone metastatic potential were observed [10,11]. Moreover, IL-8 is highly expressed in ER negative BC and HER2 positive BC, but it increases invasiveness and metastatic potential of both ER negative and ER positive BC cells [12]. Additionally, Johnstone et al. reported the correlation of IL-11 expression in BC tissue with a high histological grade along with poor patient survival [13]. Parallel and massive next-generation sequencing (NGS) markedly advanced our understanding of the genomic profiles of tumors [14]. Recently, NGS has allowed us to understand the genetically altered pathways involved in cancer development and progression. In addition, using NGS to identify specific germline or somatic mutations has been proven to have predictive and prognostic values [15]. We previously conducted targeted DNA sequencing to know how Egyptian patients differ from other populations in terms of somatic mutations. So, we developed a cohort study to explore the landscape of somatic mutations in Egyptian BC patients and identify the most frequently detected somatic mutations [16]. In another study on 58 BC patients, we identified all the somatic, non-silent, base substitution, and indel mutations per megabase of the examined genome to calculate the tumor mutation burden (TMB) and established an Egyptian TMB prediction model based on the expression level of ER, PR, HER-2, and Ki-67 [17]. In the present study, we used Ion AmpliSeq Comprehensive Cancer Panel-targeted sequencing to study the involvement of somatic mutations in interleukin signaling pathways associated with grade II invasive BC Egyptian patients to broaden our understanding of their role in promoting carcinogenesis and developing personalized therapy in Egypt.

## 2. Materials and Methods

### 2.1. Patient Samples

Fresh tissue samples were recruited from the National Cancer Institute (NCI) of Egypt from November 2019 to November 2020. Fifty-five grade II invasive BC patients were included in this study. Data for somatic mutations in 45 BC patients were already available from a previous study [17]. In addition, the study included data for ten new BC patients' somatic mutations. Our study included female patients of any age with grade II invasive BC, and tumor size less than 5 cm (T1–T2). Patients who had received neoadjuvant chemotherapy were excluded. The collected tissues were stored in MACS Tissue Storage Solution at −80 in the freezer till DNA extraction. Each patient enrolled in this study signed a written informed consent form. The clinicopathological features of the enrolled patients were collected from the clinical records at NCI, Cairo University, Cairo, Egypt.

### 2.2. DNA Extraction, Library Preparation, and Data Processing

Genomic DNA was isolated, according to the manufacturer's instructions, from fresh tissue using QIAamp DNA Mini Kit (Cat. No. 51304, Qiagen, Hilden, NRW, Germany). The isolated DNA was measured using Qubit® 3.0 Fluorometer (Cat. No. Q33216, Thermofischer Scientific Inc, Waltham, MA, USA) with Qubit™ dsDNA HS assay kit (Cat. No. Q32854, Thermofischer Scientific Inc, USA). Then, the amplicon libraries were prepared using Ion Ampliseq Complete Cancer Panel (CCP) and Ion AmpliSeq™ Library Kit Plus (Cat. No. 4488990, Life Technologies, Carlsbad, CA, USA). The CCP has 16,000 primer pairs for 409 genes (Table S1), covering 15,749 mutations. Libraries QCs were evaluated using QIAxcel (Cat No. 900194 Qiagen, Hilden, NRW, Germany). QPCR of the libraries

was performed using the Ion Library Quantitation Kit (Cat. No. 4468802, Life Technologies, Waltham, MA, USA). Ion Proton Sequencing 200 Kit v2 (Cat. No. 4485149, Life Technologies, Waltham, MA, USA) was utilized for next-generation sequencing on the Ion Proton Platform. Sequencing data were primarily analyzed using the Torrent Suite as recommended by the manufacturer. The low-quality reads <Q20 and the bases with low base quality were trimmed out. A run was accepted only if the total number of aligned reads covered at least 95% of the target regions and the depth of coverage for the variants was larger than $100\times$.

### 2.3. Identification and Classification of the Somatic Variants in Our Study

Firstly, we filtered the somatic mutations according to our normal Egyptian population database. Then, we used the Catalogue of Somatic Mutations in Cancer (COSMIC) in the identification and classification of the somatic variants. Then, we used ANNOVAR package to annotate the somatic variants. All genetic annotations and nomenclature were based on GRCh37/hg19 build. Frameshift deletions/ insertions, and in-frame deletions/insertions were considered to be pathogenic. For missense mutations, we used amino acid prediction software, SIFT [18], PolyPhen-2 [19], and CADD Phred score [20], to predict their deleteriousness. A variant was considered "damaging" when any prediction program predicted it as "damaging". According to the American College of Medical Genetics and Genomics (ACMG) recommendations, the variants were classified into the following: pathogenic variant (PV), likely pathogenic variant (LPV), variants of uncertain significance (VUS), likely benign (LB), and benign (B) [21]. The novel variants predicated to be deleterious were submitted to the ClinVar submission portal in order to obtain accession numbers for the variants (Submission ID: SUB11788117 & Organization ID: 507536; Genomic Center, National Cancer Institute, Cairo, Egypt).

### 2.4. Pathway Analysis by Ingenuity Variant Analysis (IVA)

For pathway analysis, we used ingenuity variant analysis (IVA; QIAGEN, Germany) to identify the significantly altered interleukin signaling pathways. The Ingenuity Knowledge Base was used to link variants in an analyzed dataset to biologically relevant information. Based on the applied filter, the pathogenic variants and the most affected pathways could be identified.

## 3. Results

### 3.1. Clinicopathological Characteristics of the Studied Sample Set

The clinical characteristics of the studied patients are summarized in Table 1. Patients were aged from 32 to 73 years old. According to the menopausal status, there were only 8 premenopausal patients (14.5%) and 47 postmenopausal patients (85.5%). Regarding hormonal status, 67.3% and 65.4% were positive for estrogen (ER) and progesterone (PR), respectively. However, only 21.8% of the patients were positive for HER-2, and 65.4 % tested positive for KI-67 $\geq$ 14%.

**Table 1.** Clinicopathological characteristics of the BC patients.

| Patients Characteristics | Total (*n* = 55) | Percentage (%) |
|---|---|---|
| Age: Median (Range) | 52.6 (32–73) | |
| <50 | 18 | 32.7 |
| ≥50 | 37 | 67.3 |
| Menopausal status | | |
| Premenopausal | 8 | 14.5 |
| Postmenopausal | 47 | 85.5 |

**Table 1.** *Cont.*

| Patients Characteristics | Total (*n* = 55) | Percentage (%) |
| --- | --- | --- |
| Lymph Node involvement | | |
| 0 | 12 | 21.8 |
| 1 to 3 | 22 | 40 |
| >3 | 21 | 38.2 |
| Estrogen (ER) | | |
| Negative | 18 | 32.7 |
| Positive | 37 | 67.3 |
| Progesterone (PR) | | |
| Negative | 19 | 34.5 |
| Positive | 36 | 65.4 |
| HER-2 | | |
| Negative | 43 | 78.2 |
| Positive | 12 | 21.8 |
| Ki-67 | | |
| <14% | 19 | 34.5 |
| ≥14% | 36 | 65.4 |

### 3.2. Somatic Mutations Involved in Interleukin Signaling Pathways

Without departing from the scope of this study, we reported the somatic mutations involved in the significantly altered interleukin signaling pathways. Table 2 lists the identified somatic mutations involved in interleukin signaling pathways. We identified 21 mutations in components of the interleukin-2, interleukin-6, and interleukin-7 signaling pathways, including somatic mutations in *JAK1*, *JAK2*, *SOCS1*, *IL7R*, *MCL1*, *BCL2*, *MTOR*, and *IL6ST* genes. Interestingly we identified 18 out of 21 variants that were not previously reported in BC tissue. We identified four loss-of-function mutations in the *JAK1* gene in 13 (23.6%), 8 (14.5%), 5 (9%), and 3 (5.4%) patients, respectively. This gene harbored four different mutations in 29 patients, representing 52.7%. The *BCL2* gene harbored two different deletions in 8 patients (16.4%). The *MCL1* gene harbored three different mutations in 11 (20%), one (1.8%), and one (1.8%) patients, respectively. In addition, two different mutations were identified in the *SOCS* gene and *IL7R* gene in 15 (27.3%) and 10 (18.2%) patients, respectively. One loss function mutation was identified in each of the *JAK2* gene, *IL6ST* gene, and *MTOR* gene.

### 3.3. The Identified Likely Novel Mutations of Somatic Allele Origin

Interestingly, two likely novel mutations predicted to be deleterious were identified in the *SCH1* gene (p.K11fs*18 and p.F17fs*12); each has been identified in two samples. Two more likely novel mutations were identified in the *IL-2* gene (p.C145fs*8 in 10 patients and p.L45* in 3 patients). In addition, one likely novel mutation was identified in each of the *IL-7R* gene (p.K157fs*4) and *JUN* gene (p.P220fs*6), as listed in Table 3. The raw data for the identified somatic mutations in our study are illustrated in Table S2.

**Table 2.** Somatic mutations identified in our multigene panel screening.

| Gene | Position | Cosmic ID | Type | IVA Classification | Inferred Activity | SIFT/or PolyPhen-2 Prediction | CADD Phred Score | Affected Transcript | HGVS.c | HGVS.p | Cases (*n* = 55) | Occurrence in Cosmic Database |
|------|----------|-----------|------|--------------------|--------------------|-------------------------------|------------------|---------------------|--------|--------|------------------|-------------------------------|
| *AKT3* | Chr1:243675627 | COSM7264038 * | Deletion | PV | Loss | - | 32 | NM_001370074.1 | c.1353delA | p.K451fs*? | 9 (16.3%) | Large intestine = 1 |
| *AKT3* | Chr1:243708813 | COSM8568034 * | Deletion | PV | Loss | - | - | NM_001370074.1 | c.1250del | p.K417Sfs*10 | 3 (5.4%) | Stomach = 1 |
| *BCL2* | Chr18:60985665 | COSM7343104 * | Deletion | PV | Loss | - | - | NM_000633.2 | c.234_235del | p.G79Rfs*73 | 1 (1.8%) | Large intestine = 1 |
| *BCL2* | Chr18:60985776 | COSM9113142 * | Deletion | PV | Loss | - | - | NM_000633.2 | c.124delG | p.A42fs*54 | 7 (12.7%) | Large intestine = 1 |
| *IL6ST* | Chr5:55247869 | COSM269367 * | Deletion | PV | Loss | - | 32 | NM_175767.3 | c.1587delA | p.V530* | 14 (25.4%) | Large intestine = 12 and Stomach = 1 |
| *IL7R* | Chr5:35867541 | COSM2687237 * | Deletion | PV | Loss | - | 20.2 | NM_002185.5 | c.361delA | p.I121* | 7 (12.7%) | Large intestine = 7 and Stomach = 4 |
| *IL7R* | Chr5:35873646 | COSM7342389 * | SNV | PV | Loss | Damaging | 25.3 | NM_002185.5 | c.602A > G | p.Y201C | 3 (5.4%) | Large intestine = 1 and Thyroid = 1 |
| *JAK1* | Chr1:65325833 | COSM1560531 | Deletion | PV | Loss | - | 33 | NM_001321853.2 | c.1289delC | p.P386fs*2 | 13 (23.6%) | Large intestine = 8 and Breast = 3 |
| *JAK1* | Chr1:65330630 | COSM1343915 * | Deletion | PV | Loss | - | 27.9 | NM_001321853.2 | c.884delA | p.N339fs*3 | 8 (14.5%) | Endometrium = 2 and Large intestine = 1 |
| *JAK1* | Chr1:65339129 | COSM6912578 * | Deletion | PV | Loss | - | 32 | NM_001321853.2 | c.407delA | p.N136fs*32 | 5 (9%) | Prostate = 2 |
| *JAK1* | Chr1:65339111 | COSM1639943 | Deletion | PV | Loss | - | 25.6 | NM_001321853.2 | c.425delA | p.K142fs*26 | 3 (5.4%) | Breast = 4 and Large intestine = 2 |
| *JAK2* | Chr9:5078342 | COSM7286153 * | Deletion | PV | Loss | - | 33 | NM_001322199.1 | c.1586delA | p.N678fs*53 | 10 (18%) | Large intestine = 1 |
| *JAK3* | Chr19:17954215 | COSM34216 * | SNV | PV | Gain | Possibly damaging | 19.01 | NM_000215.4 | c.394C>A | p.P132T | 3 (5.4%) | Lung = 1 and Hematopoietic and lymphoid = 1 |
| *MCL1* | Chr1:150551952 | COSM5156964 * | Deletion | PV | Loss | - | 22.7 | NM_001197320.1 | c.55delG | p.A19fs*30 | 11 (20%) | Upper aerodigestive tract = 1 |
| *MCL1* | Chr1:150551935 | COSM5734360 * | Deletion | PV | Loss | - | 15.72 | NM_001197320.1 | c.61_72delTTGGCGCCCGG | p.C21CG24del | 1 (1.8%) | Large intestine = 1 and Stomach = 1 |
| *MCL1* | Chr1:150551327 | COSM6923093 * | SNV | NV | Loss | Tolerated | 22.8 | NM_001197320.1 | c.680C>T | p.A227V | 1 (1.8%) | Thyroid = 1 and Hematopoietic and lymphoid = 1 |

**Table 2.** *Cont.*

| Gene | Position | Cosmic ID | Type | IVA Classification | Inferred Activity | SIFT/or PolyPhen-2 Prediction | CADD Phred Score | Affected Transcript | HGVS.c | HGVS.p | Cases (*n* = 55) | Occurrence in Cosmic Database |
|---|---|---|---|---|---|---|---|---|---|---|---|---|
| *MTOR* | Chr1:11199444 | COSM6716875 * | SNV | PV | Loss | Damaging | 32 | NM_004958.4 | c.5047C>T | p.R1683W | 1 (1.8%) | Large intestine = 1 |
| *NRAS* | Chr1:115252309 | COSM7177458 * | SNV | PV | Loss | Possibly Damaging | 23.4 | NM_002524.5 | c.331A>G | p.M111V | 1 (1.8%) | Urinary tract = 1 and Stomach = 1 |
| *PIK3CA* | Chr3:178936082 | COSM29329 | SNV | PV | Loss | Damaging | 31 | NM_006218.4 | c.1624G>A | p.E542K | 11 (20%) | Breast = 239 and Large intestine = 66 |
| *SOCS1* | Chr16:11348846 | COSM19320 * | Deletion | PV | Loss | - | 24.3 | NM_003745.1 | c.490delG | p.A164fs*41 | 10 (18%) | Hematopoietic and lymphoid = 3 and Large intestine = 1 |
| *SOCS1* | Chr16:11348998 | COSM6964419 * | Deletion | PV | Loss | - | 32 | NM_003745.1 | c.338delT | p.F113fs*5 | 5 (9%) | Large intestine = 1 |

HGVS.c: Human Genome Variation Society, Coding DNA sequence; HGVS.p: Human Genome Variation Society, protein sequence; Chr.: Chromosome.; BC: breast cancer. Variants were classified for their pathogenicity according to ClinVar and FATHMM (#) predictions; PV: pathogenic variants; NV: neutral variant. (*) indicates variants appeared in other tissue rather than BC.

**Table 3.** Novel mutations identified in our multigene panel screening.

| Gene | Position | Cosmic ID | Accession No. | Type | IVA Classification | Inferred Activity | Affected Transcript | HGVS.c | HGVS.p | Cases (n = 55) |
|---|---|---|---|---|---|---|---|---|---|---|
| *SHC1* | Chr1:154947252 | Novel | SCV002574988 | Deletion | PV | Loss | NM_001130041.2 | c.33delA | p.K11fs*18 | 2 (3.6%) |
| *SHC1* | Chr1:154947270 | Novel | SCV002574989 | Deletion | PV | Loss | NM_001130041.2 | c.51delT | p.F17fs*12 | 2 (3.6%) |
| *IL2* | Chr4:123372936 | Novel | SCV002574990 | Deletion | PV | Loss | NM_000586.4 | c.433delT | p.C145fs*8 | 10 (18%) |
| *IL2* | Chr4:123377462 | Novel | SCV002574991 | Deletion | PV | Loss | NM_000586.4 | c.134delT | p.L45* | 3 (5.3%) |
| *IL7R* | Chr5:35871245 | Novel | SCV002574992 | Deletion | PV | Loss | NM_002185.5 | c.470delA | p.K157fs*4 | 5 (9%) |
| *JUN* | Chr1:59248084 | Novel | SCV002574993 | Deletion | PV | Loss | NM_002228.4 | c.659delC | p.P220fs*6 | 7 (12.7%) |

HGVS.c: Human Genome Variation Society, Coding DNA sequence; HGVS.p: Human Genome Variation Society, protein sequence; Chr.: chromosome.; BC: breast cancer. Variants were classified for their pathogenicity according to ClinVar and FATHMM (#) predictions; PV: pathogenic variants.

### 3.4. The Most Commonly Altered Pathways

Pathway analysis was performed using ingenuity variant analysis (IVA, QIAGEN, Germany). Pathway Analysis reported the altered interleukin signaling pathways with its involved mutated genes. The most altered interleukin signaling pathways were IL-2 ($p$ = 0.0021) (Figure 1), IL-6 ($p$-value = 0.0025) (Figure 2), and IL-7 ($p$-value = 0.0012) (Figure 3), signaling pathways that are altered in 34.5%, 29%, and 23.6% of patients, respectively.

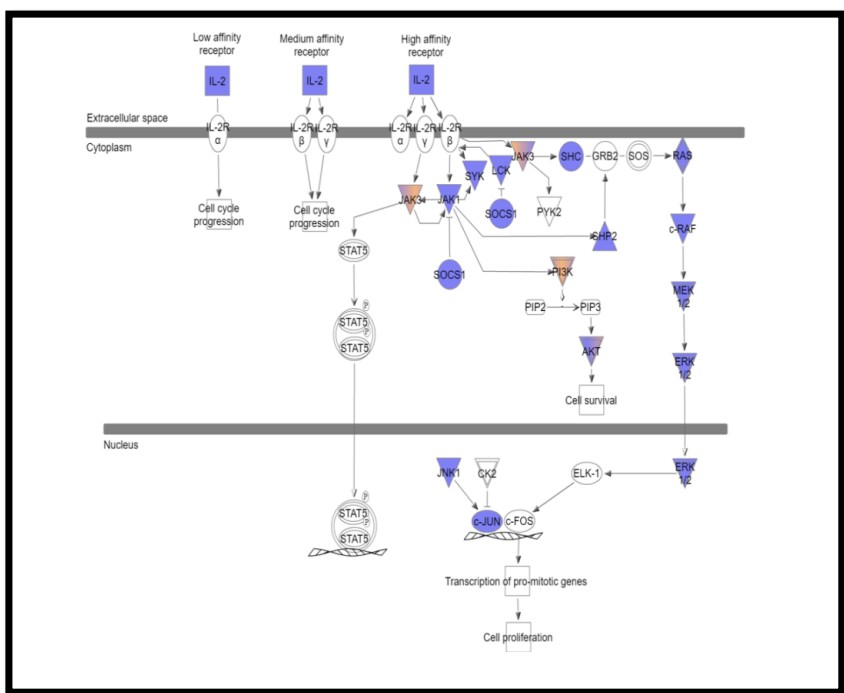

**Figure 1.** IL-2 signaling pathway identified using ingenuity variant analysis (IVA). Blue represents loss of function, and orange represents gain of function.

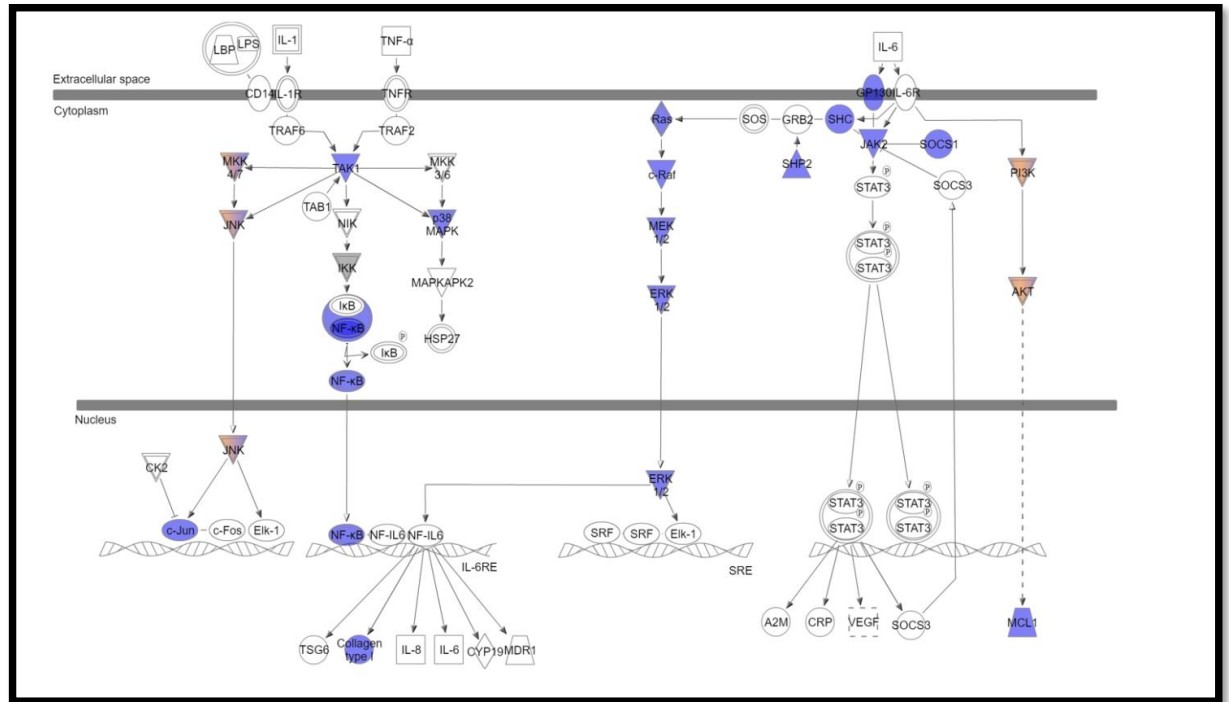

**Figure 2.** IL-6 signaling pathway identified using ingenuity variant analysis (IVA). Blue represents loss of function, and orange represents gain of function.

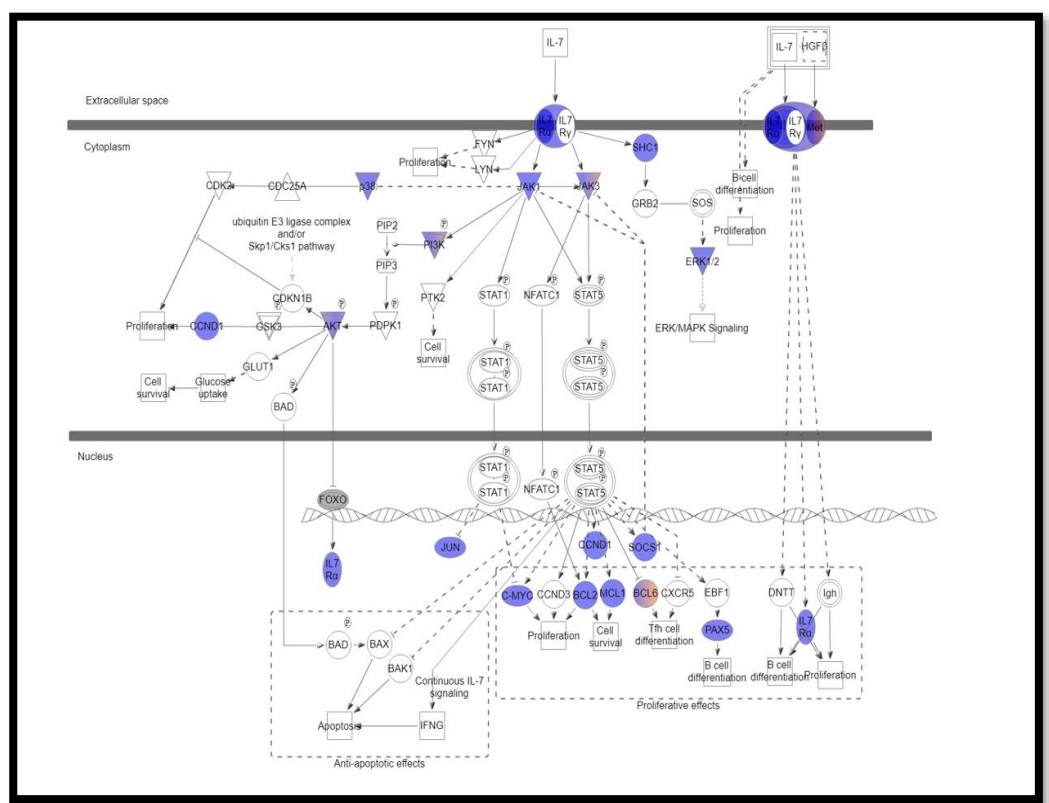

**Figure 3.** IL-7 signaling pathway identified using ingenuity variant analysis (IVA). Blue represents loss of function, and orange represents gain of function.

## 4. Discussion

Cancers, including BC, are multifactorial diseases in which genetic and epigenetic changes significantly impact tumorigenesis, progression, and treatment response [22]. These alterations in normal cells lead to dysregulation in many signaling pathways, including interleukin signaling pathways. Interleukins serve as a means of communication between immune and non-immune cells and tissues [23]. Therefore, interleukins are essential in cancer development and progression by nurturing an environment that promotes cancer growth [24]. Understanding different pathways that underlie BC development and progression is critical to identifying new targeted therapeutic agents and is relevant to the disease diagnosis and prognosis [25]. In the present study, somatic mutations involved in various interleukin signaling pathways were identified in fifty-five Egyptian grade II invasive BC patients. Our results showed that all patients have at least one somatic mutation in ILs-signaling pathways. According to Ingenuity Variant/Pathway Analysis, the most altered interleukin signaling pathways in our study were interleukin-2, interleukin-6, interleukin-7, and interleukin-15 signaling pathways.

IL-2 is a key regulator of normal immune functions and is critical for the activation and subsequent amplification of the immune response following antigenic stimulation [26]. IL-2 binds with high affinity to the trimeric IL-2 receptor (IL-2R), consisting of three chains (IL-2Rα, β, and γ). IL-2R uses Janus family kinase (JAK) members to induce signal transduction [27,28]. The IL-2Rβ chain binds to JAK1, and the IL-3Rγ binds to JAK3, which will result in JAK1/3 activation and activate the IL-2-mediated proliferative responses [29]. We identified loss of function mutations in *JAK1* and gain of function mutations in *JAK3* genes and hypothesized that it resulted in constitutive activation of their kinase activity. From here comes the importance of using next-generation sequencing techniques to identify the mutations likely to lead to constitutive activation of kinases. Such mutations are known to be a driving event in cancer, and their identification is of great importance as it might help identify their selective kinase inhibitors. Consequently, we

claim that JAK1 and/or JAK3 selective kinase inhibitors can be used to block IL-2-mediated proliferative responses [30,31].

Moreover, we identified loss of function mutations in the *SHC* gene, the key pathway for IL-2 control of Ras/Raf/MEK/REK pathway. Tyrosine-phosphorylated SHC forms a complex with the SH2 domain of the adapter GRB2. This complex then binds to Ras guanine nucleotide exchange protein SOS (Son of Sevenless) and activates the Raf-ERK MAP kinase cascade [32]. The referred loss of function mutations are novel and are reported in our study for the first time.

IL-2 can also regulate the activation of the PI3K-AKT signaling pathway [33,34] through the recruitment of the regulatory subunit of PI3K (p85 subunit) into the IL-2Rβ signaling complex [35]. Interestingly, the recruitment of this regulatory subunit to the receptor is mediated by JAK1 through a non-phosphotyrosine interaction between it and IL-2Rβ [36]. Therefore, mutations in the *JAK1* gene might explain why the PI3K-AKT signaling pathway is upregulated in our studied cases.

Several studies suggested the relationship between inflammation and various cancer types and showed that anti-inflammatory agents could attenuate tumor growth. Interleukin-6 (IL-6) is a pro-inflammatory cytokine released by cancerous cells in the tumor microenvironment and plays a vital role in the differentiation and expansion of tumor cells [37,38]. The increased and dysregulated activities of IL-6 can be found in cancer. Additionally, the mutations in the gene-encoding proteins involved in the IL6-activated intracellular signaling cascades make these proteins constitutively active, resulting in hyperactivation of these signaling pathways. Thus, these mutations are interesting targets for IL6-signaling targeted cancer therapy, which aims to block the activation of pathways downstream of IL-6 and reduce the proliferation and survival of cancer cells [39].

IL-6 leads to the activation of three major downstream pathways due to the dimerization of gp130/IL6ST: the JAK/STAT pathway, the MAPK/MERK/ERK signaling cascade, and the PI3K/AKT pathway. The tyrosine kinases in the JAK family that are associated with the cytoplasmic region of gp130/IL6ST by a non-covalent bond, initiate the signaling cascade. The dimerization of gp130/IL6ST causes auto-phosphorylation and activation of JAK. In turn, JAK activates the SH2 domain-containing cytoplasmic protein tyrosine phosphatase (SHP2), which activates the Ras/Raf/MEK/ERK pathway, responsible for different transcription factors linked to cell growth [40,41]. Moreover, JAK can also activate the PI3K/AKT pathway. The activation of these three main signaling pathways leads to tumor-promoting cytokines' effect on both cancer cell-intrinsic processes, such as cell proliferation, differentiation, survival, invasion, and metastasis, and cancer cell-extrinsic processes, such as modulation of inflammation and angiogenesis and the effect on the tumor microenvironment (TME) [42,43]. Here, in our study, we identified somatic mutations in genes controlling the IL-6 signaling pathway, loss of function mutations in IL-6 signal transducer (IL6ST; gp130), loss of function mutations in *JAK1*, gain of function mutation in *JAK3*, and loss of function mutation in *SOCS1*. Thus, we hypothesized that *JAK1* mutations lead to a partial loss of its kinase activity. On the other hand, all the previously mentioned mutations promote the constitutive activation of the PI3K/AKT pathway and Ras/Raf/MEK/ERK pathway. We also identified mutations in the *SOCS1* gene, the negative feedback gene of these signaling pathways.

IL-6 is mainly secreted in the tumor cells and tumor associated fibroblast and many studies have reported its immunopathogenicity and its signaling in tumor growth, metastasis, and therapeutic resistance in BC. However, herein, we shed light on IL-6 function in grade II invasive BC patients. Therefore, targeting IL-6 and/or its receptor in combination with other potent anticancer therapies may be a potent therapeutic approach for BC therapy [44–46].

IL-7/IL-7R signaling activates two main signal transduction pathways (the JAK/STAT and PI3K/Akt pathway). The first mandatory step in IL-7/IL-7R signaling is the phosphorylation of JAK1 and JAK3 kinases, which will result in constitutive activation of proteins regulating cell cycle progression, survival, and protein translation [47]. Additionally, IL-7

stimulation will induce the JAK-mediated activation of STAT5, facilitating its nuclear translocation and regulation of gene expression [48]. Herein, we claimed that mutations in *IL7R*, *JAK1*, and *JAK3* would result in constitutive activation of the IL7R-JAK-STAT signaling pathway. In addition, we hypothesized that loss-of-function mutations in *CCND1* and *SOCS1* genes cause deregulation of expression of antiapoptotic (Bcl-2 and Mcl-1) and proapoptotic (Bad and Bax) proteins.

These findings shed light on the alteration of IL-signaling pathways in our BC patients. These results were in agreement with our previous published study in Egyptian colorectal cancer patients that reported the alteration of the interleukin pathways in those patients [49], suggesting that Egyptian patients are likely to be prone to these alterations. Therefore, these data highlight the prominent role of ILs as promising candidates for targeted therapy in Egyptian patients.

## 5. Conclusions

In summary, our findings support the role of somatic mutations in IL-signaling pathways in breast oncogenesis in Egyptian patients with grade II BC.

Using multigene panel sequencing analysis, we were able to identify somatic mutations in IL-signaling pathways including *JAK1, JAK2, SOCS1, IL7R, MCL1, BCL2, MTOR,* and *IL6ST* genes. Six mutations are likely to be novel and deleterious. In addition to the diagnostic benefit of detecting such IL-pathways alterations, this work would also assist in the development of personalized treatment regimens suited to Egyptian breast cancer patients. Further research is recommended to validate these identified somatic mutations in larger series of patients and to confirm their impact at the protein level.

## 6. Recommendation

We recommend following up the response to treatment of those patients to investigate the better described therapy regimens that are suitable for the identified mutations in this study.

**Supplementary Materials:** The following supporting information can be downloaded at: https://www.mdpi.com/article/10.3390/cimb44120401/s1, Table S1: The Comprehensive Cancer Panel (CCP) gene list, Table S2: The raw data for the identified somatic mutations in our study.

**Author Contributions:** A.R.N.Z. (dead) and A.N.: conceptualization; A.N., M.H.E., M.M.L. and A.S.E.-D.Y.: methodology and software; A.M.L.: resources; A.N., A.R.N.Z. and A.S.E.-D.Y.: investigation and validation; A.N., A.R.N.Z. and M.A.: data curation and formal analysis; A.R.N.Z.: supervision and funding acquisition; A.N.: writing—original draft; M.H.E. and A.S.E.-D.Y.: review and editing. All authors have read and agreed to the published version of the manuscript.

**Funding:** The current study is funded by the Science and Technology Development Fund (STDF), Project ID: 41907.

**Institutional Review Board Statement:** The study protocol was accepted by the Institutional Review Board (IRB number: IRB00004025; approval number: 2010014038.3) of the National Cancer Institute (NCI), Cairo University, Egypt. The study was guided according to the ICH-GCP guidelines and a written informed consent was received from each patient enrolled in the study.

**Informed Consent Statement:** Informed consent was obtained from all subjects involved in the study.

**Data Availability Statement:** The data generated during this study are included in this article while, its Supplementary Information files are available upon request.

**Acknowledgments:** The authors acknowledge the enrolled patients from the NCI for the fresh tissues provided for research.

**Conflicts of Interest:** The authors do not have any competing interests.

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
