# Peer review of "Somatic Mutations Alter Interleukin Signaling Pathways in Grade II Invasive Breast Cancer Patients: An Egyptian Experience"

_cimb, doi:10.3390/cimb44120401_

Round 1

Reviewer 1 Report

The authors investigated the occurence of somatic mutations on interleukine and signaling pathways in grade II breast cancer patients (n=55) in Egypt. Although they detected several ILs (IL-2, IL-6 and IL7) I somehow miss IL-1, IL-8, IL-11, IL-12 changes which are well known to have functions in breast cancer occurance as well as progression.

They detected a loss of function of AKT3- which is known to be associated with ER regative status.

We find the sentence: "In 2020, WHO reported 2.3 million women diagnosed with BC and 68,000 deaths worldwide" on the first page- 68000 must be changed to 680000.

Reviewer 2 Report

In this study, the authors described the impact of somatic mutations on various interleukin signaling pathways associated with grade II invasive breast cancer (BC) in Egyptian patients with the aim to provide more insights in the process of carcinogenesis.

Main points:

- In the Introduction section, the authors could better describe the background and rationale of this study. The authors could better describe the investigation of involvement of somatic mutations in interleukin signaling pathways associated with invasive BC and reporting the most recent literature.

- The study needs further evaluation to better describe the therapy regimens suitable for specific target mutations of BC cases described in this study.

- In the Section 2.3, the authors should clarify the use of COSMIC database in the identification and classification of the somatic variants and add the annotation variant tool used in this study, after the sequencing data processing.

- In the method section, the authors could add the description of method used for pathway analysis (Ingenuity Variant Analysis -IVA) for the identification of the most significantly altered pathways, whose results are reported in the specific section of the results (3.2).

Minor points:

- In the Section 3.1, the authors could use the same rounding for all percentage values. In this section, the authors also should check the age range in the text (line 113) in comparison with the age range indicated in the Table 1 (the range seems to be different in the text and in the table of Clinicopathological characteristics of the BC patients).

- In the Section 3.2, the authors could better organize this section. Specifically, the authors could use the same type of description used for mutations in JAK1 gene for all genes mentioned, including (or excluding) the HGVS nomenclature for all variants.

In the text, the authors reported three different deletions (line 130) identified in the BCL2 gene, but in the table 2 only two different deletions were reported for the same gene.

In addition, the authors could use the same criteria for the description of genes in the text and in table 2 (in alphabetic order or by the number of deletions).

Round 2

Reviewer 1 Report

I don't see too much changes between the old and new versions.

Round 3

Reviewer 1 Report

I accept the changes.